# Plant Hormone and Inorganic Ion Concentrations in the Xylem Exudate of Grafted Plants Depend on the Scion–Rootstock Combination

**DOI:** 10.3390/plants11192594

**Published:** 2022-10-01

**Authors:** Kohei Kawaguchi, Makoto Nakaune, Jian Feng Ma, Mikiko Kojima, Yumiko Takebayashi, Hitoshi Sakakibara, Shungo Otagaki, Shogo Matsumoto, Katsuhiro Shiratake

**Affiliations:** 1Graduate School of Bioagricultural Sciences, Nagoya University, Chikusa, Nagoya 464-8601, Japan; 2Saitama Agricultural Technology Research Center, Sugahiro, Kumagaya 360-0102, Japan; 3Research Institute for Bioresources, Okayama University, Chuo, Kurashiki 710-0046, Japan; 4RIKEN Center for Sustainable Resource Science, Tsurumi, Yokohama 230-0045, Japan

**Keywords:** tomato, eggplant, grafting, xylem exudate, ionome analysis, hormonome analysis

## Abstract

In grafted plants, inorganic ions and plant hormones in the xylem exudate transported from the rootstock to the scion directly or indirectly affect the scion, thereby improving the traits. Therefore, the concentration of these components in the xylem exudate of grafted plants may be an indicator for rootstock selection. On the other hand, few reports have presented a comprehensive analysis of substances transferred from the rootstock to the scion in plants grafted onto different rootstocks, primarily commercial cultivars. In this study, we measured inorganic ions and plant hormones in the xylem exudate from the rootstock to the scion in various grafted plants of tomato and eggplant. The results revealed that the concentrations of inorganic ions and plant hormones in the xylem exudate significantly differed depending on the type of rootstock. In addition, we confirmed the concentration of the inorganic ions and plant hormones in the xylem exudate of plants grafted onto the same tomato rootstock cultivars as rootstock with tomato or eggplant as the scions. As a result, the concentrations of inorganic ions and plant hormones in the xylem exudate were significantly different in the grafted plants with eggplant compared with tomato as the scion. These results suggest that signals from the scion (shoot) control the inorganic ions and plant hormones transported from the rootstock (root).

## 1. Introduction

In plants, the xylem transports various long-distance signaling molecules that mediate root-to-shoot communication. The xylem exudate contains water and inorganic ions absorbed by the roots from the soil [1] and small molecules such as sugars, amino acids, organic acids, plant hormones, and peptides, as well as polymeric compounds such as proteins and polysaccharides [2], and these long-distance signaling molecules alter the shoot traits.

For instance, water and salt stress in tomato cultivation suppresses water uptake by roots and transport to the shoot, increasing starch accumulation in fruits at an early developmental stage [3] and inducing the concentration effect of metabolites in fruits at a later developmental stage [4], leading to increased fruit sugar content.

Inorganic ions transported from roots to shoots alter plant traits. Zn plays an essential role in various processes such as nucleic acid synthesis, gene expression regulation, the removal of reactive oxygen species (ROS), and plant hormone synthesis [5,6,7,8]. Excess Zn causes decreased yield, poor growth, chlorosis due to impaired chlorophyll synthesis and chloroplast degradation, and inhibition of the absorption of other essential elements, such as Ca, Mg, Fe, and P [6,9,10,11]. In plants, Ca acts as a cell wall and plasma membrane stabilizer, enzyme activator, and intracellular signal [12]. Ca deficiency inhibits root and pollen tube elongation and causes leaf chlorosis, resulting in stunted plant growth [13]. In horticultural crops, Ca affects fruit quality by contributing to physiological disorders such as bitter pit in apples [14] and blossom-end rot in tomatoes [15].

Cytokinins (CKs) that are biosynthesized primarily in the roots and are transported to the shoots through the xylem influence various characteristics of plant growth, development, and physiology, such as leaf senescence, apical dominance, seed germination, and flower and fruit development. Mutants of genes involved in CK signaling or biosynthesis in *Arabidopsis* exhibit shoot growth retardation, including reduced rosette leaf, scale, and number of flower buds [16,17].

Small peptide molecules secreted in the xylem play essential roles in cell-to-cell and organ-to-organ signaling in plant morphogenesis and environmental responses [18,19]. The C-terminally encoded peptide (CEP) is a N starvation-induced small 15-amino-acid peptide hormone, which acts as a root-derived ascending N-demand signal to the shoot through the xylem [20]. CEP is recognized in the shoot by the specific leucine-rich repeats (LRR) receptor kinase CEP reporter 1 (CEPR1), which induces the production of non-secreted polypeptides CEP downstream 1 and 2 (CEPD 1 and 2) as a secondary signal [21]. CEPD 1 and 2 acts as a shoot-derived descending signal to the root via the phloem and ultimately upregulates the gene expression of the nitrate transporter *NRT2.1* in non-deprived roots [21].

Grafting is a common horticultural technique in which plant tissues from different plants are joined together to form a new composite plant [22] and is widely used in the commercial production of Cucurbitaceae crops, such as watermelon (*Citrullus lanatus*), melon (*Cucumis melo*) and cucumber (*C. sativus*), and Solanaceae crops, such as tomato (*Solanum lycopersicum*), eggplant (*S. melongena*), and pepper (*Capsicum annuum*) [23,24], as well as the large-scale vegetative propagation of desirable fruit trees [25]. Grafting provides additional advantages to new composite plants, such as improved tolerance to abiotic stresses [24,26,27] and increased nutrient uptake and utilization efficiency in several plant species [28,29]. Furthermore, the use of a specific rootstock in grafting affects the amount of substances transported through the xylem exudate and alters the traits of the scion. The details of this process are well researched and are described below.

Excess B in the soil or irrigation water severely compromises plant growth and yield. Grafting the susceptible tomato cultivar ‘Ikram’ onto the commercial rootstock ‘Arnold’ (an inter-specific hybrid between *S. lycopersicum* and *S. habrochaites*) reduced the expression of genes encoding B transporters such as SlNIP5 and SlBOR1 in the root of the graft and improved its tolerance to excess B compared to that in self-grafted and non-grafted plants [30]. The bloom is a white fine powder on the surface of cucumber fruits that is primarily composed of silica (SiO_2_) [31]. In the late 1980s, cucumbers without bloom (bloomless cucumbers) became popular in Japan because of their attractive and distinctly shiny appearance. Previous studies have shown that a single amino acid mutation in the silicon (Si) influx transporter (CmLsi1) of the bloomless pumpkin rootstock affects its localization in the plasma membrane, resulting in a loss of Si transport from an external solution to the root cells [32]. Potassium (K) is the major cation in ripe grapes, and grape juice pH is strongly correlated with its K content. Thus, K also determines the pH of wine after fermentation, with higher wine pH negatively affecting wine color, stability, and taste [33]. Rootstocks control the concentration of K in the scion, and the accumulation of K in the whole grapevine or shoot is positively related to the concentration of K in the xylem exudate [34].

Cucumber grafted onto pumpkin rootstock increased the abscisic acid (ABA) concentration in the xylem exudate from the rootstock to the scion, mediating rapid stomatal closure in the cucumber scion, and improved osmotic stress tolerance under salinity stress conditions [35]. Apple *(Malus domestica*) grafted onto the dwarfing ‘M.9’ rootstock reduced gibberellin and CK synthesis in the root and their transport to the shoot [36] and significantly decreased the mean total shoot length and node number of the scion compared to ‘MM.106’, ‘M.793’, and ‘Royal Gala’ rootstocks [37]. Sweet cherry (*Prunus avium* L.) grafted onto six different rootstocks exhibited significant fruit set and yield [38]. Additionally, the simultaneous increase in the CK concentration in the xylem exudate from the rootstock with fruit set and yield suggests that its transport from the rootstock may regulate these characteristics [38]. Watermelon grafted onto squash (*C. maxima* × *C. moschata* ‘Shintosa’) exhibited more vigorous growth than a non-grafted plant, and its CK activity, primarily ribosyl zeatin content, in the xylem exudate was significantly higher than that in the non-grafted plants [39]. Peach (*Prunus persica* L. ‘Batsch’) was grafted onto three *Prunus* species characterized by different levels of growth vigor, and the hormone transport rates in the xylem exudate from both the grafted plants and the non-grafted plants were compared [40]. Growth potential and zeatin riboside levels were positively correlated, whereas a negative correlation was found with indoleacetic acid levels in the xylem exudate of peaches grafted onto the same rootstocks. Therefore, the differences in the growth vigor of the grafted plants might be explained by the decreased polar translocation of indole-3-acetic acid (IAA) due to increased transport of CKs to the shoot from the root [40].

Thus, current research indicates that inorganic ions and plant hormones in the xylem exudate transported from the rootstock to the scion directly or indirectly affect the rootstock, thereby improving the traits of the scion in grafted plants. Therefore, the concentration of these components in the xylem exudate of grafted plants may be an indicator for rootstock selection. On the other hand, few reports have presented a comprehensive analysis of substances transferred from the rootstock to the scion in plants grafted onto different rootstocks, primarily commercial cultivars. Therefore, this study comprehensively analyzed the inorganic ions and plant hormones in the xylem exudate transfer from the rootstock to the scion in grafted tomato and eggplant commercial cultivars. The results revealed that the concentrations of inorganic ions and plant hormones in the xylem exudate differed significantly depending on the rootstock type. This information can be used to select rootstocks for specific purposes such as inducing abiotic stress tolerance, improving fruit quality, and controlling growth vigor. Furthermore, the concentrations of inorganic ions and types of CK molecules in the xylem exudate from the rootstock were regulated by the scion, suggesting that there may be a mechanism for the uptake of inorganic ions and biosynthesis or transfer of CK molecules tailored to the characteristics of the shoot.

## 2. Results and Discussion

In this study, xylem exudates from various grafted tomato plants and eggplants were sampled and analyzed to determine the inorganic ions and plant hormones transferred from the rootstock to the scion. At the end of the cultivation (7–8 months after grafting), xylem exudates were collected for 1 h from the main stem of the scion at 10 cm from the grafted position for the analysis of the inorganic ion and plant hormone concentrations.

### 2.1. Inorganic Ion Concentrations in the Xylem Exudate of Tomato and Eggplant Grafted onto Various Rootstocks

When tomato ‘CF Momotaro Haruka’ as a scion was grafted onto various tomato cultivars (rootstock cultivars; ‘Ganbarune’, ‘Ganbarune-Triper’, ‘Spike’, dwarf cultivars; ‘Chibikko’, ‘Micro-Tom’, ‘Regina’) or wild tomato species (*S. pennellii*) as rootstock (Appendix A) and eggplant ‘Shikibu’ as a scion was grafted onto various tomato rootstock cultivars (‘Ganbarune’, ‘Ganbarune-Triper’, ‘Spike’) or eggplant rootstock cultivars (‘Daitaro’, ‘Hiranasu’, ‘Tonashim’) as rootstocks (Appendix A), there was no difference in the concentrations of K, Mg, Cu, and Ni in the xylem exudate of the grafted plants of tomato or eggplant compared to their concentrations in the respective self-grafted plants (Figure 1A,D and Figure 2D,F).

#### 2.1.1. Ca Concentration in the Xylem Exudates of Plants Grafted onto Various Rootstocks

The Ca concentration in the xylem exudates of plants with tomato as the scion grafted onto ‘Spike’ and *S. penellii* as rootstocks was significantly higher and tended to be higher in the plants with tomato as the scion grafted onto ‘Ganbarune’ and ‘Ganbarune-Tripar’ as rootstocks compared to that in the self-grafted plants (Figure 1B). The Ca concentration in the xylem exudate of plants with eggplant as the scion grafted onto tomato ‘Ganbarane’ and ‘Spike’ as rootstocks tended to be higher, but it tended to be lower in plants grafted onto eggplant ‘Hiranasu’ as rootstock compared to that in the self-grafted plants (Figure 1B).

In plants, Ca acts as a stabilizer of the cell wall and plasma membrane, enzyme activator, and intracellular signal [12]. In general, Ca deficiency in plants inhibits root and pollen tube elongation and causes the chlorosis of leaves, leading to stunted plant growth [13]. Moreover, in horticultural crops, Ca causes physiological disorders, such as bitter pit in apples, affecting fruit quality [14]. Blossom-end rot is an important physiological disorder in tomato, and its incidence is negatively correlated with the soluble Ca concentration in the fruit [22,41]. It is conjectured that inadequate Ca supply in the fruit apoplast leads to a Ca-deficient state and disrupts the homeostatic system of the cell wall and plasma membrane structure, thereby inducing blossom-end rot [42]. The reason why the Ca concentration tended to be higher in the xylem exudates of plants with tomato as the scion grafted onto tomato rootstock cultivars (‘Ganbarane’, ‘Ganbarune-Triper’, and ‘Spike’) as rootstocks compared to that in self-grafted plants may be due to the empirical selection of rootstocks with high Ca supply from root to shoot to prevent tomato fruit blossom-end rot.

#### 2.1.2. P Concentration in the Xylem Exudates of Plants Grafted onto Various Rootstocks

The P concentration in the xylem exudate of plants with tomato as the scion grafted onto ‘Spike’ as rootstock was significantly higher and also exhibited a higher trend in the plants with tomato as the scion grafted onto ‘Ganbarune’, ‘Ganbarume-Tripar’, and ‘Micro-Tom’ as rootstocks compared to that in the self-grafted plants (Figure 1C). The P concentration in the xylem exudate of plants with eggplant as the scion grafted onto eggplant ‘Hiranasu’ as the rootstock tended to be lower than that in the self-grafted plants (Figure 1C).

P is an essential nutrient for plant growth, a component of proteins, nucleic acids, and lipid membranes, and plays a vital role in nitrogen metabolism, carbohydrate transport, and carbohydrate and fat metabolism in plants [43]. P-deficient plants exhibit delayed root development [44] and decreased fruit yield and quality [45].

P is applied to farmland in the form of P fertilizers for crop production; however, the inorganic phosphorus rock used as a raw material for fertilizers is a finite resource expected to be depleted within a few centuries [46]. Moreover, the majority of the P applied to farmland is either lost as runoff from the farmland, adsorbed to the soil, or metabolized to organic compounds and therefore its uptake by plants is hindered, owing to which it accumulates in the soil. Thus, there is a need to breed plants that use P efficiently. This study revealed that the P concentration in the xylem exudate of plants with tomato as the scion grafted onto ‘Spike’ as the rootstock was significantly higher and tended to be higher in plants with tomato as the scion grafted onto ‘Ganbarune’, ‘Ganbarune-Tripar’, and ‘Micto-Tom’ as rootstocks compared to that in the self-grafted plants (Figure 1C). Therefore, the use of these tomato cultivars as rootstocks may improve the efficiency of grafted plants to use the limited phosphorus resources.

#### 2.1.3. Fe Concentration in the Xylem Exudates of Plants Grafted onto Various Rootstocks

The Fe concentration was significantly lower in the xylem exudates of plants with tomato as the scion grafted onto ‘Ganbarune-Triper’, ‘Chibikko’, ‘Micro-Tom’, ‘Regina’, and *S. penellii* as rootstocks compared to that in the self-grafted plants (Figure 1E). The Fe concentration in the xylem exudate of plants with eggplant as the scion grafted onto eggplant ‘Tonashim’ as the rootstock tended to be higher, but in the plants grafted onto eggplant ‘Hiranasu’ as the rootstock, it tended to be lower compared to that in the self-grafted plants (Figure 1E).

Fe plays an essential role in various processes such as the electron transfer in respiration, photosynthesis, and enzymatic reactions such as redox and DNA synthesis in plants [47]. In Fe-deficient plants, leaves near the actively growing shoot apical meristem become chlorotic, leading to poor plant growth [47,48].

#### 2.1.4. Mn Concentration in the Xylem Exudates of Plants Grafted onto Various Rootstocks

The Mn concentration in the xylem exudate of plants with tomato as the scion grafted onto *S. penellii* as the rootstock was significantly lower and tended to be lower in the plants with tomato as the scion grafted onto ‘Micro-Tom’ as the rootstock than that in the self-grafted plants (Figure 2A). The Mn concentration in the xylem exudate of plants with eggplant as the scion grafted onto tomato ‘Ganbarune’, ‘Spike’, and eggplant ‘Tonashim’ as rootstocks tended to be higher compared to that in the self-grafted plants (Figure 2A).

Mn plays essential roles in various processes, such as photosynthesis, respiration, ROS removal, disease resistance, and plant hormone signaling in plants [49]. Mn-deficient plants have decreased photosynthetic efficiency due to the decreased chloroplast numbers and chlorophyll content [50,51,52], resulting in reduced biomass [53,54,55]. In contrast, Mn excess in plants induces chlorosis in leaves [56] due to the inhibition of chlorophyll biosynthesis [57,58], which decreases the photosynthetic rate [59,60] and inhibits auxin biosynthesis, which decreases cell division in root meristematic tissues [61] and inhibits the absorption and transport of other essential elements such as Ca, Mg, Fe, and P [62,63,64,65].

#### 2.1.5. Zn Concentration in the Xylem Exudates of Plants Grafted onto Various Rootstocks

The Zn concentration in the xylem exudate of plants with tomato as the scion grafted onto ‘Micro-Tom’ as the rootstock was significantly lower but it tended to be higher in grafted plants with tomato as the scion grafted onto *S. penellii* as the rootstock than that in the self-grafted plants (Figure 2B). The Mn concentration in the xylem exudate of plants with eggplant as the scion grafted onto tomato ‘Ganbarune’, ‘Spike’, and eggplant ‘Tonashim’ as rootstocks tended to be higher compared to that in the self-grafted plants (Figure 2B).

Zn plays an essential role in various processes such as nucleic acid synthesis, gene expression regulation, ROS removal, and the synthesis of plant hormones [5,6,7,8]. Excess Zn causes decreased yield, poor growth, chlorosis due to decreased chlorophyll synthesis and chloroplast degradation, and inhibition of the absorption of other essential elements, such as Ca, Mg, Fe, and P [5,9,10,11].

#### 2.1.6. B Concentration in the Xylem Exudates of Plants Grafted onto Various Rootstocks

The B concentration in the xylem exudate of plants with tomato as the scion grafted onto *S. penellii* as the rootstock was significantly lower than that in the self-grafted plants (Figure 2C). The B concentration in the xylem exudate of plants with eggplant as the scion grafted onto tomato ‘Ganbarune’ and eggplant ‘Daitaro’ as rootstocks tended to be higher than that in the self-grafted plants (Figure 2C).

B plays an essential role in the ester cross-linking of rhamnogalacturonan II in the cell wall and contributes to maintaining cell strength and adhesion through the formation of pectin networks in plants [66]. B deficiency inhibits root cell elongation and enlargement, reduces young leaf development due to decreased cell division, and decreases fertility by inhibiting pollen tube elongation [67]. On the other hand, excess B results in decreased root cell division, leaf chlorophyll content, photosynthetic rate, and lignin and suberin contents [68], as well as inhibition of shoot and root growth [69].

B is transported by boric acid channels, such as AtNIP5;1 [70], which are responsible for boric acid uptake from the soil into the root cells, and B transporters, such as AtBOR1 [71], which transport boric acid from the cytosol to the extracellular space in *Arabidopsis*. Tomato plants with ‘Ikram’ scion grafted onto the commercial rootstock ‘Arnold’ (an inter-specific hybrid between *S. lycopersicum* and *S. habrochaites*) exhibited improved tolerance to excess B compared to the self-grafted and non-grafted plants by reducing the expression of genes encoding B transporters such as SlNIP5 and SlBOR1 in the roots [30]. Therefore, these boric acid channels and B transporters may be responsible for the differences in the B concentrations in the xylem exudates of grafted plants with various tomato and eggplant cultivars as rootstocks.

#### 2.1.7. Mo Concentration in the Xylem Exudates of Plants Grafted onto Various Rootstocks

The Mo concentration in the xylem exudate of plants with tomato as the scion grafted onto ‘Chibikko’ and ‘Regina’ as rootstocks was significantly higher compared to that in the self-grafted plants (Figure 2E). The B concentration in the xylem exudate of plants with eggplant as the scion grafted onto eggplant ‘Tonashim’ as the rootstock tended to be higher compared to that in the self-grafted plants (Figure 2E).

Mo plays an essential role in the metabolism of nitrogen, purine, and plant hormone biosynthesis in plants [72,73]. Mo-deficient plants had altered morphology of leaves, the poor development of seeds, impairment of flower production, and decreased overall plant growth [72].

In this study, the concentration of inorganic ions in the xylem exudates of plants of tomato and eggplant scions grafted onto various rootstocks showed that the concentration of inorganic ions in the xylem exudate of grafted plants differed significantly depending on the type of rootstock, such as different cultivars or species.

Previous studies have selected rootstocks that promote the uptake of inorganic ions into plants to improve nitrogen uptake efficiency [74] and also selected rootstocks that inhibited the uptake of inorganic ions into the plants for improved resistance to excess B, Cu, Mn, and Cd [28,75,76,77]. Furthermore, previous studies have reported that the concentration of inorganic ions in the fruit correlated positively with the concentration of inorganic ions in the xylem exudate in grafted plants [78]. Therefore, information on the concentration of inorganic ions in the xylem exudate of grafted plants in this study may contribute to the better selection of rootstocks that are resistant to the excess or deficiency of inorganic ions or may increase or decrease the concentration of inorganic ions in the fruit during tomato and eggplant cultivation.

### 2.2. Plant Hormone Concentrations in the Xylem Exudates of Tomato and Eggplant Plants Grafted on Various Rootstocks

When tomato ‘CF Momotaro Haruka’ as the scion was grafted onto various tomato cultivars or wild tomato species as rootstocks (Appendix A) and eggplant ‘Shikibu’ as the scion was grafted onto various tomato or eggplant cultivars as rootstocks (Appendix A), auxin and gibberellin were undetectable in the xylem exudates. The salicylic acid concentration in the xylem exudate of grafted plants showed no difference compared to the concentration in the self-grafted plants (Appendix A).

#### 2.2.1. ABA Concentration in the Xylem Exudates of Plants Grafted onto Various Rootstocks

The ABA concentration in the xylem exudates of plants with tomato as the scion grafted onto ‘Spike’ and ‘Chibikko’ as rootstocks was significantly lower but tended to be higher in the plants with tomato as the scion grafted onto ‘Micro-Tom’ and *S. penellii* as rootstocks compared to that in the self-grafted plants (Figure 3A). The ABA concentration in the xylem exudate of plants with eggplant as the scion grafted onto eggplant ‘Tonashim’ as the rootstock was significantly higher compared to that in the self-grafted plants (Figure 3A).

ABA contributes to tolerance to various abiotic stresses, such as drought, salt, and high and low temperatures [79]. Roots that sense drought stress promote the biosynthesis of ABA [80], which is transported to the leaves through xylem exudates [81,82,83,84]. The resultant effects of stomatal closure suppress transpiration losses, presumably leading to drought stress tolerance.

The results suggest that ‘Micro-Tom’ and *S. pennellii* as rootstocks in tomato-grafted plants and ‘Tonashim’ as the rootstock in eggplant-grafted plants may have sensed drought stress due to an imbalance between transpiration from the leaves of the scion and water absorption by the roots of the rootstock. In addition, a decrease in fruit size and an increase in sugar content have been observed in tomato plants grafted onto ‘Micro-Tom’ as the rootstock (Nakaune et al., unpublished). These characteristics are observed in high-sugar tomato cultivation under drought stress [85,86], supporting the hypothesis that ABA concentrations in the xylem exudate increased due to the drought stress condition sensed by tomato plants grafted onto ‘Micro-Tom’ rootstock. Thus, grafted plants with high ABA concentrations in the xylem exudate may induce tolerance to abiotic stresses such as drought and cold stress.

#### 2.2.2. JA Concentration in the Xylem Exudates of Plants Grafted onto Various Rootstocks

Jasmonic acid (JA) participates in fruit ripening, growth, development, and senescence [87,88,89] and contributes to biotic stress tolerance, such as herbivore and disease resistance, in addition to abiotic stress tolerance [90,91].

The JA concentration in the xylem exudate of plants with tomato as the scion grafted onto ‘Micro-Tom’ and *S. penellii* as rootstocks was significantly higher than that in the self-grafted plants (Figure 3B). Moreover, the JA concentration in the xylem exudate of grafted plants with the scion, such as tomato, had a positive correlation (R = 0.76) with the ABA concentration (Figure 3). This result is consistent with previous studies showing that JA and ABA mutually regulate each other’s biosynthesis [92,93,94,95], and mutants of JA biosynthesis or signaling pathways alter water stress tolerance [96,97,98]. The analysis of grafted plants of the *def-1* mutant, which does not accumulate JA in dry soils, revealed that JA accumulates not only in local tissues and organs but is also transported long-distance via the transpiration channel from the root to the shoot to regulate transpiration [94]. The results suggest that the high concentration of JA in the xylem exudate of tomato plants grafted onto ‘Micro-Tom’ as a rootstock may affect water and drought stress tolerance.

In this study, the analysis of the xylem exudates of plants of tomato and eggplant scions grafted on various rootstocks showed that the ABA and JA concentrations varied depending on the rootstock. In addition, tomato plants grafted onto ‘Micro-Tom’ as a rootstock with higher ABA and JA concentrations in the xylem exudate (Figure 3) exhibited an increased fruit sugar content, and a decreased fruit size and yield (Nakaune et al., unpublish). These results suggest that ABA and JA concentrations in the xylem exudate may be indicators of the fruit sugar content and yield.

#### 2.2.3. CK Concentrations in the Xylem Exudates of Plants Grafted onto Various Rootstocks

CKs are classified into four types: N6-(Δ 2-isopentenyl)-adenine (iP), tZ (trans-zeatin), cis-zeatin (cZ), and dihydrozeatin (DZ) [99]. tZ-type CKs are primarily synthesized in roots and transported to the shoot via an apoplastic pathway to promote plant growth [100,101]. In contrast, iP-type CKs are the primary forms found in the phloem sap, which act as a soil nitrogen signal, decreasing nitrogen uptake [102] and regulating root development by modulating polar auxin transport and vascular bundle formation in the root meristem [103].

The concentrations of cZ- and DZ-type CKs in the xylem exudates of tomato or eggplant grafted onto any of the tomato or eggplant cultivars as rootstocks were less than 3 pmol/mL (Appendix A). The analysis of iP- and tZ-type CKs in the xylem exudates of plants with tomato or eggplant as the scion grafted onto any tomato or eggplant as the rootstock revealed that the concentrations of iPRPs, iP9G, tZ9G, tZOG, and tZRPsOG were less than 0.1 pmol/mL or below the detection limit (Appendix A), and the concentrations of tZRPs and tZROG were less than 0.5 pmol/mL (Appendix A).

The iP concentration in the xylem exudates of plants with tomato as the scion grafted onto ‘Ganbarune-Triper’, ‘Chibikko’, ‘Regina’, Micro-Tom’, and *S. pennellii* as rootstocks was significantly lower than that in the self-grafted plants (Figure 4A). The iP concentration in the xylem exudates of plants with eggplant as the scion grafted onto eggplant ‘Daitaro’ and ‘Tonashim’ as rootstocks tended to be higher compared to that in the self-grafted plants (Figure 4A).

The iPR concentration in the xylem exudate of plants with tomato as the scion grafted onto ‘Ganbarune-Triper’, ‘Chibikko’, ‘Micro-Tom’, and *S. pennellii* as rootstocks was significantly lower compared to that in the self-grafted plants (Figure 4B). The iPR concentration in the xylem exudate of plants with eggplant as the scion grafted onto eggplant ‘Daitaro’ and ‘Tonashim’ and tomato tended to be higher compared to the self-grafted plants (Figure 4B).

The iP7G concentration in the xylem exudate of tomato or eggplant grafted onto any tomato or eggplant rootstocks was not significantly different compared to the respective self-grafted plants (Figure 4C).

The tZ concentration in the xylem exudates of plants with tomato as the scion grafted onto *S. pennellii* as the rootstock was significantly lower and it tended to be lower in the plants grafted onto ‘Ganbarune-Triper’ and ‘Regina’ as rootstocks compared to the self-grafted plants (Figure 4D). The tZ concentration in the xylem exudate of plants with eggplant as the scion grafted onto tomato as rootstocks tended to be lower, but the plant grafted onto eggplant as the rootstock exhibited a higher tZ concentration trend than that in the self-grafted plants (Figure 4D).

The tZR concentration in the xylem exudate of plants with tomato as the scion grafted onto ‘Ganbarune’, ‘Ganbarune-Triper’, ‘Spike’, ‘Micro-Tom’, and *S. pennellii* as rootstocks was significantly lower than that in the self-grafted plants (Figure 4E). The tZR concentration in the xylem exudate of plants with eggplant as the scion grafted onto tomato ‘Ganbarune-Triper’ and ‘Spike’ as rootstocks was significantly lower but tended to be higher in the plants grafted onto eggplant ‘Daitaro’ and ‘Tonashim’ as rootstocks compared to the self-grafted plants (Figure 4E).

The tZ7G concentration in the xylem exudate of plants with tomato or eggplant as the scion grafted onto any tomato or eggplant as rootstocks was not significantly different compared to that of the self-grafted plants (Figure 4F). The tZ7G concentration in the xylem exudate of plants with eggplant as the scion grafted onto tomato ‘Ganbarune’, ‘Ganbarune-Triper’, and eggplant ‘Tonashim’ as rootstocks was significantly lower compared to the self-grafted plants (Figure 4F).

CKs play essential roles in a variety of processes, such as cell division and differentiation, seed germination, apical bud dominance, leaf senescence, root growth, and stress responses [99,104,105,106]. Watermelon plants grafted onto pumpkin as the rootstock showed higher CK activity in the xylem exudate and more vigorous growth than non-grafted plants [39]. This suggests that the CK concentration in the xylem exudate depends on the type of rootstock, resulting in altered yield and vigor of the plant.

iP and tZ, the main CK active forms in plants, are catalyzed by iPRPs and tZRPs, respectively, by the CK-activating enzyme LONELY GUY (LOG) [107,108]. In addition, the ABCG14 transporter plays a role in loading tZ-type CKs into xylem [109,110]. Therefore, the enzyme that catalyzes the biosynthesis of CKs and CK transporters may be responsible for the differences in the CKs concentration in the xylem exudate of plants grafted onto various tomato and eggplant rootstocks.

Although CK activity is thought to be regulated by the molecular type, the analysis of CYP735A, which catalyzes the biosynthesis of iP- to tZ-type CKs, demonstrated that only tZ-type CKs are involved in shoot growth [17]. Furthermore, tZ transported from roots through the xylem exudate regulates leaf size; however, tZ biosynthesized from tZR by LOG in the shoot regulates not only leaf size but also leaf formation. This suggests that not only the quantity but also the molecular type and transported form of CK are important for plants. Therefore, the molecular types of CKs in the xylem exudate may be altered by the rootstock cultivar, thereby affecting the traits of the grafted plants.

### 2.3. Scions Alter Inorganic Ion Concentrations and CK Molecular Types Transported from the Rootstock to the Scion

In this study, we analyzed inorganic ions and plant hormones in the xylem exudate of plants grafted onto tomato rootstock cultivars (‘Ganbarune’, ‘Ganbarune-Triper’, ‘Spike’) as rootstocks with tomato ‘CF Momotaro Haruka’ or eggplant ‘Shikibu’ as the scions. Initially, we expected that the concentration of inorganic ions in the xylem exudates would not significantly differ between tomato and eggplant as the scions when grafted on the same tomato cultivar as the rootstock. However, the concentrations of inorganic ions such as Ca, P, Mg, Fe, Mn, Zn, Cu, and Mo in the xylem exudates were significantly lower in the grafted plant with eggplant as the scion than with tomato as the scion grafted onto the same tomato cultivar as the rootstock (Figure 1 and Figure 2). The concentrations of all inorganic ions in the xylem exudate were not significantly different for the same tomato cultivar as the rootstock compared to the self-grafted eggplants (Figure 1 and Figure 2). In addition, the concentrations of K and Ni in the xylem exudate that were not significantly different between self-grafted eggplants and self-grafted tomato plants were not significantly different between grafted plants with eggplant and tomato as the scions grafted onto the same tomato cultivar as the rootstock. These results suggest that the tomato scion and the tomato rootstock can communicate, but the eggplant scion and the tomato rootstock cannot communicate or the eggplant scion controls inorganic ion uptake of the tomato rootstock. CEP is transported from the root to the shoot via the xylem and is recognized by CEPR1, which produces CEPD1 and 2 as secondary signals [21]. CEPD1 and 2 are transported from the shoot to the root via the phloem and ultimately upregulate the expression of the nitrate transporter *NRT2.1* gene in non-depleted roots [21]. This study suggests that signals that regulate the expression of transporter genes, such as CEPD1 and 2, from the scion (shoot) control the transport of inorganic ions or uptake by the rootstock (root).

tZ-type CKs are synthesized mainly in the roots and transported to the shoot to promote growth [100,101]. Therefore, we expected that the percentage of CK molecules in the xylem exudate would not be significantly different between plants with tomato and eggplant as the scions grafted onto the same tomato cultivar as the rootstock. The percentage of tZ-type CKs in the xylem exudate was 65.1% to 69.4% with tomato as the scion, whereas it was 29.0% to 40.0% with eggplant as the scion, on the same tomato cultivar as the rootstock (Figure 5). The percentage of iP-type CKs in the xylem exudate was 26.8% to 31.1% with tomato as the scion, whereas it was 55.6% to 67.5% with eggplant as the scion, on the same tomato cultivar as the rootstock. In the xylem exudate of tomato self-grafted plants, the percentage of tZ-type CKs was 66.9% and the percentage of iP-type CKs was 29.8%. In contrast, in the xylem exudate of plants with eggplant as the scion grafted onto eggplant cultivar as the rootstock, the percentage of tZ-type CKs was 51.4% to 53.9%, and the percentage of iP-type CKs was 37.9% to 47.1%. The types of CK molecules in the xylem exudate of plants grafted onto tomato rootstock cultivar with tomato as the scion were similar to those in the tomato self-grafted plants, but in grafted plants with eggplant, were similar to self-grafted eggplants. These results suggest that unknown signals from the scion (shoot) control the molecular type of CKs transported from the rootstock (root) or biosynthesized by the rootstock (root).

Previous studies have suggested that some signals from the shoot control CKs that are transported from the root to the shoot. In pea (*Pisum sativum*) *rms* mutants, zeatin riboside in the xylem sap decreased (1/40th times) compared with that in the wild type (WT) [100]. The amount of zeatin riboside in the xylem sap of plants grafted onto the *rms* mutant as the rootstock and WT as the scion increased compared to that in the self-grafted *rms* mutant [111]. In contrast, the amount of zeatin riboside in the xylem sap of plants with the *rms* mutant as the scion grafted onto the WT rootstock decreased compared to the self-grafted plant of the *rms* mutant [111]. This also suggests that a signal from the shoot controls CK transport from the root to the shoot. However, this signaling molecule has not yet been identified. In this study, although the signaling molecule was not identified, the findings strongly support the hypothesis that the presence of a signal regulates CK transfer from the root to the shoot.

## 3. Materials and Methods

### 3.1. Plant Materials and Growth Condition

As scions, tomato ‘CF Momotaro Haruka’ (Takii Seed, Kyoto, Japan) and eggplant ‘Shikibu’ (Watanabe Seed, Miyagi, Japan) were used. Three rootstock tomato cultivars, ‘Ganbarune’, ‘Ganbarune-Triper’, ‘Spike’ (Aisan Seed, Aichi, Japan), three dwarf tomato cultivars, ‘Chibikko’ (Marutane, Kyoto, Japan), ‘Micro-Tom’ (MT-J, NBRP), ‘Regina’ (Sakata seed, Kanagawa, Japan), one wild tomato *S. pennellii* (NBRP), and three eggplant rootstock cultivars, ‘Daitaro’, ‘Hiranasu’ (*S. aethiopicum*), and ‘Tonashim’ (*S. torvum*) (Takii Seed, Kyoto, Japan) were used (Appendix A).

Cleft grafting was performed for the *S. pennellii* rootstock and splice grafting for the other rootstocks [23]. The grafted plants were grown under conventional cultivation using non-woven pots in the greenhouse of the Saitama Prefectural Agricultural Technology Center (Saitama, Japan).

### 3.2. Collection of Xylem Exudates from the Main Stem

At the end of cultivation (7–8 months after grafting), the main stem of the scion at 10 cm from the grafted position was cut, and the xylem exudate was collected in a tube for 1 h.

### 3.3. Ionome Analysis

The measurement of inorganic ion concentrations was performed according to Jing et al. (2019) [112]. Briefly, the xylem exudate was freeze-dried and then digested in 60% (*w*/*w*) HNO3 at a temperature up to 140 °C. Water was added to the digested sample and centrifuged. The supernatant was diluted two times with 5% (*w*/*w*) HNO3. Metal concentrations were determined by inductively coupled plasma/mass spectrometry (ICP/MS, 7700X, Agilent Technologies, Santa Clara, CA, USA).

### 3.4. Hormonome Analysis

For the UPLC-ESI-qMS/MS analysis of each plant hormone, the xylem exudate was freeze-dried and then subjected to extraction and fractionation of plant hormones according to Kojima et al. (2009) [113] and Kojima and Sakakibara (2012) [114]. CKs were quantified by an ultra-performance liquid chromatography-electrospray interface and tandem quadrupole mass spectrometer (AQUITY UPLC System/Xevo-TQS, Waters, Milford, MA, USA) as described by Kojima et al. (2009) [113]. ABA, IAA, jasmonic acid, salicylic acid, and gibberellic acid were quantified by an ultra-high-performance liquid chromatography electrospray interface and quadrupole-orbitrap mass spectrometer (UHPLC/Q-Exactive, Thermo Scientific, Waltham, MA, USA) as described by Kojima and Sakakibara (2012) [114] and Shinozaki et al. (2015) [115].

## Figures and Tables

**Figure 1 plants-11-02594-f001:**
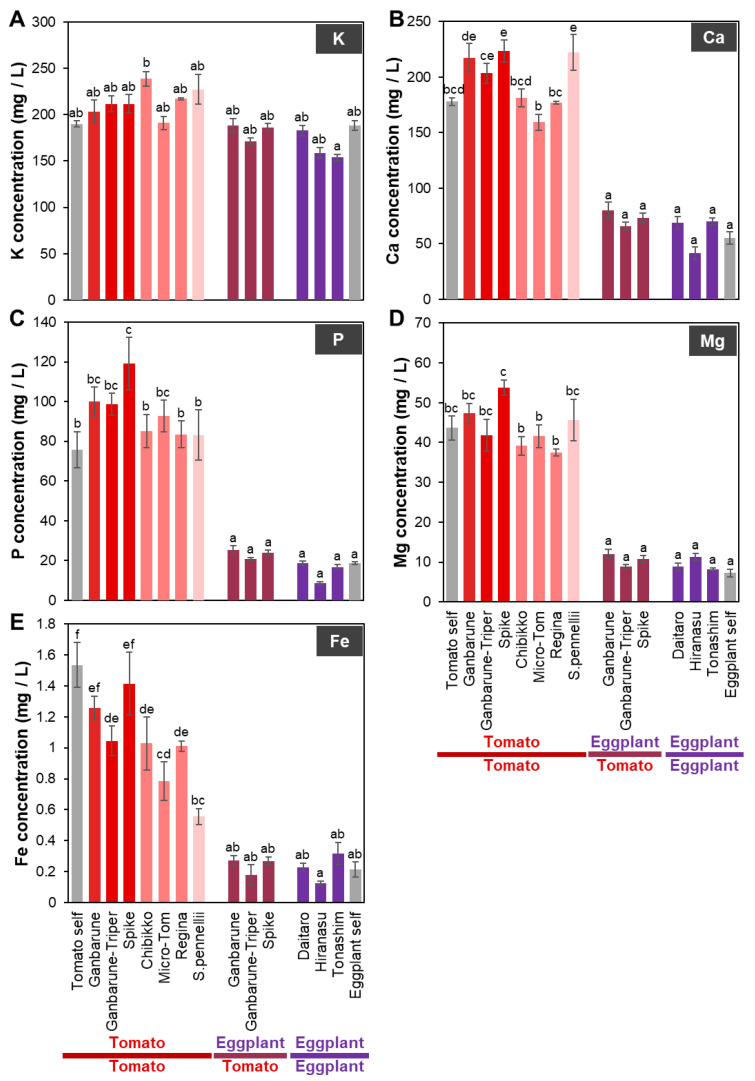
The concentrations of macroelements in the xylem exudate of various grafting combinations. The concentrations of K (**A**), Ca (**B**), P (**C**), Mg (**D**), and Fe (**E**) in the xylem exudate of various grafting combinations. Gray, red, red–pink, pink, red–purple, and purple indicate self-grafted plants, tomato grafted onto various rootstocks of tomato cultivars, tomato grafted onto various dwarf tomato cultivars, tomato grafted onto wild tomato, eggplant grafted onto various tomato cultivars, and eggplants grafted onto various eggplant cultivars, respectively. Different letters indicate significant differences according to the Tukey-Kramer test (*p* ≤ 0.05). Values are the means of four replicate samples, and error bars indicate the standard error of four replicate samples.

**Figure 2 plants-11-02594-f002:**
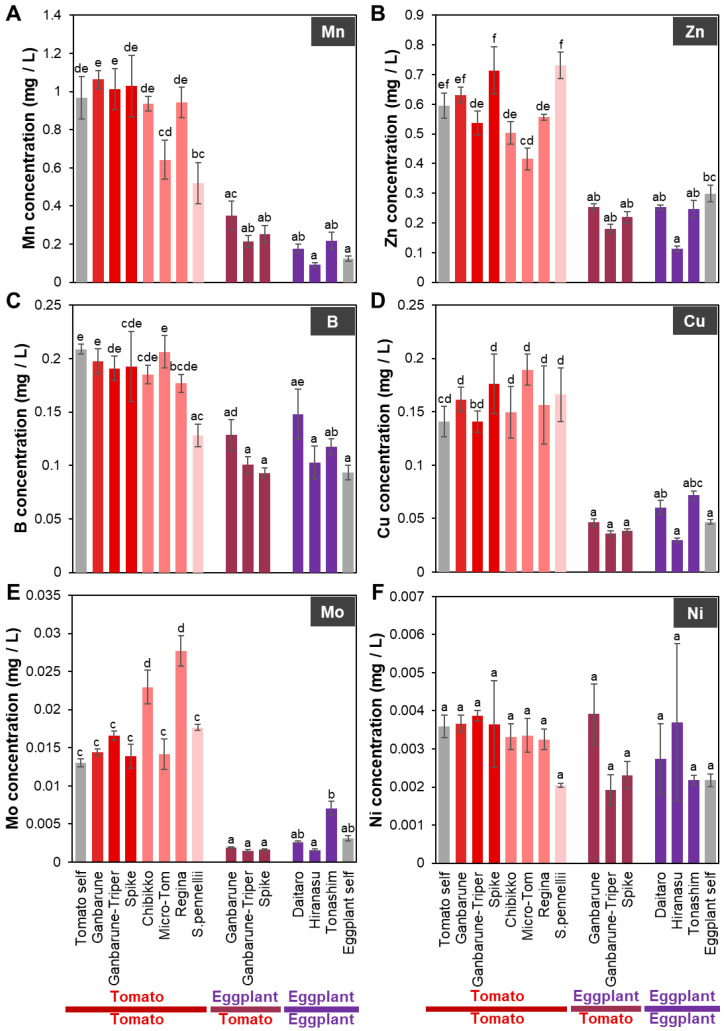
The concentrations of microelements in the xylem exudate of various grafting combinations. The concentrations of Mn (**A**), Zn (**B**), B (**C**), Cu (**D**), Mo (**E**), and Ni (**F**) in the xylem exudate of various grafting combinations. Gray, red, red–pink, pink, red–purple, and purple indicate self-grafted plants, tomato grafted onto various rootstocks of tomato cultivars, tomato grafted onto various dwarf tomato cultivars, tomato grafted onto wild tomato, eggplant grafted onto various tomato cultivars, and eggplants grafted onto various eggplant cultivars, respectively. Different letters indicate significant differences according to the Tukey-Kramer test (*p* ≤ 0.05). Values are the means of four replicate samples, and error bars indicate the standard error of four replicate samples.

**Figure 3 plants-11-02594-f003:**
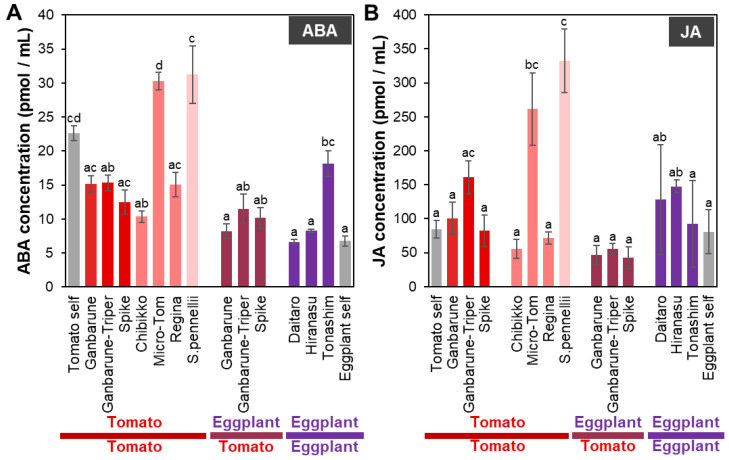
The concentrations of ABA and JA in the xylem exudates of various grafting combinations. The concentrations of ABA (**A**) and JA (**B**) in the xylem exudates of various grafting combinations. Gray, red, red–pink, pink, red–purple, and purple indicate self-grafted plants, tomato grafted onto various rootstocks of tomato cultivars, tomato grafted onto various dwarf tomato cultivars, tomato grafted onto wild tomato, eggplant grafted onto various tomato cultivars, and eggplants grafted onto various eggplant cultivars, respectively. Different letters indicate significant differences according to the Tukey-Kramer test (*p* ≤ 0.05). Values are the means of four replicate samples, and error bars indicate the standard error of four replicate samples.

**Figure 4 plants-11-02594-f004:**
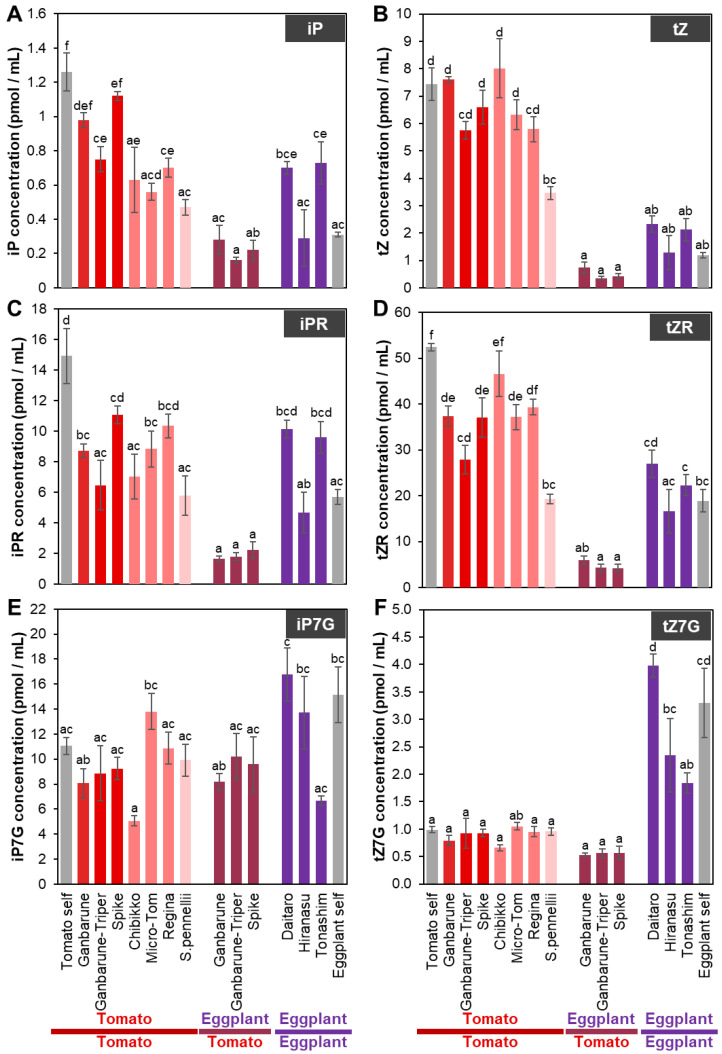
The concentration of CKs in the xylem exudates of various grafting combinations. The concentrations of N 6-(Δ 2-isopentenyl)-adenine (iP) (**A**), iP riboside (iPR) (**B**), iP-7-N-glucoside (iP7G) (**C**), trans-zeatin (tZ) (**D**), tZ riboside (tZR) (**E**), and tZ-7-N-glucoside (tZ7G) (**F**) in the xylem exudates of various grafting combinations. Gray, red, red–pink, pink, red–purple, and purple indicate self-grafted plants, tomato grafted onto various rootstocks of tomato cultivars, tomato grafted onto various dwarf tomato cultivars, tomato grafted onto wild tomato, eggplant grafted onto various tomato cultivars, and eggplants grafted onto various eggplant cultivars, respectively. Different letters indicate significant differences according to the Tukey-Kramer test (*p* ≤ 0.05). Values are the means of four replicate samples, and error bars indicate the standard error of the four replicate samples.

**Figure 5 plants-11-02594-f005:**
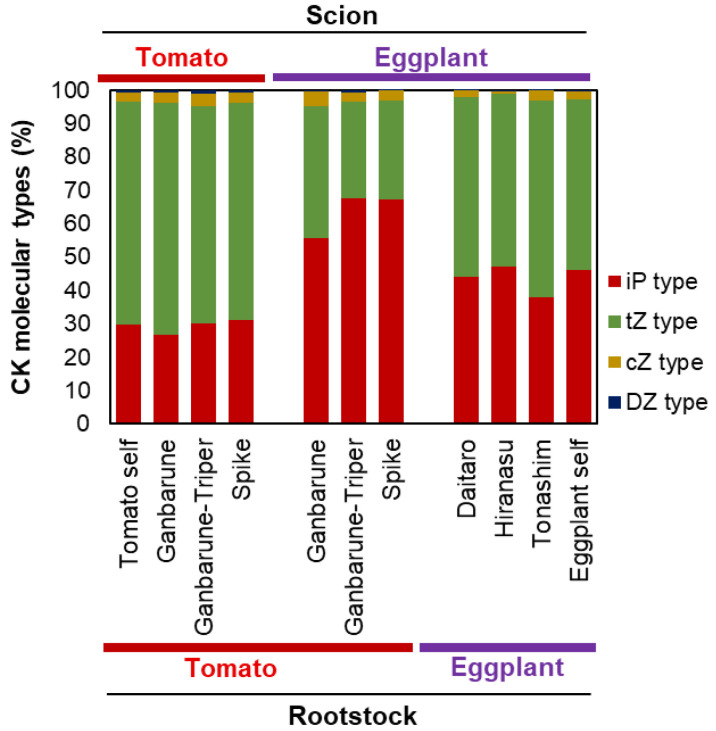
The percentage of CK molecular types in the xylem exudates of various grafting combinations. The concentrations of N 6-(Δ 2-isopentenyl)-adenine type (iP-type), trans-zeatin type (tZ-type), cis-zeatin type (cZ-type), and dihydrozeatin type (DZ-type) CKs in the xylem exudates of various grafting combinations. Red, green, yellow, and blue indicate iP type, tZ type, cZ type, and DZ type CKs, respectively. Tomato self indicates ‘CF Momotaro Haruka’ grafted onto the ‘CF Momotaro Haruka’ plant, and the eggplant self indicates ‘Shikibu’ grafted onto the ‘Shikibu’ plant. Values are the means of four replicate samples.

## Data Availability

Data can be requested from the corresponding authors.

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
