# Peer review of "Plant Hormone and Inorganic Ion Concentrations in the Xylem Exudate of Grafted Plants Depend on the Scion–Rootstock Combination"

_plants, 2022, doi:10.3390/plants11192594_

Round 1

Reviewer 1 Report

In this manuscript written by Kawaguchi et al., the authors studied on how xylem exudates are affected by rootstock genotype. Using a variety of tomato and eggplant grafts, ionome and hormonome analyses were conducted on xylem exudates from the scion stem above the graft junction. Interestingly, by exchanging rootstock genotype, a part of ion and phytohormone contents were changed dramatically. Through these observations, the authors propose that scion shoot interact and regulate rootstock status. The authors describe results obtained by previous studies and discuss the potential effect of each ion and hormone to the physiology of grafts. Overall, this study provides valuable information to consider grafting skills. I only provide minor comments.

Lines 130, “various grafted plants” Need to specify; (for examples) various tomato and eggplant grafts.

Lines 135, As Materials and Methods start below, the site where xylem exudates were collected needs to be clarified here. That information is critical to deal with all data in this study.

Lines 140-142, Only data of Cu shows no difference. This part needs to clarify.

Reviewer 2 Report

This study evaluated the ion and plant hormone contents in the xylem extracts of same tomato/eggplants scion grafted on different tomato/eggplants rootstocks. It was shown here that rootstocks influence several ion and plant hormone significantly. This data provided valuable information for tomato/eggplants rootstocks selection.

Comments:

1.       The title may need to be re-considered. The current title  scion controls ion and plant hormone contents may need to tune down. The paper, however, described the impacts of rootstocks by using the same scion grafted on different rootstocks.

2.       In line 368, it was suggested that JA concentration is positively correlated with ABA concentration. Although it seems true as the histogram shown, a correlation test will make it more clear/believable and easily to understand.

3.       The ion and plant hormone data change presented here is interesting. The authors need to provide the phenotype data, like plant architecture, fruit quality of grafted plants.

Reviewer 3 Report

Authors have analysed the concentration of inorganic ions and hormones in the xylem exudate of tomato and eggplant scions (one scion per species) grafted onto 7 tomato and 3 eggplant rootstocks. For the interspecific grafting, authors grafted the eggplant scion onto 3 selected tomato rootstocks (Ganbarune, Ganbarune-Triper and Spike). 

General comment:

I have the general sensation that the authors are presenting results from the individual experiments without putting some effort into combining results from experiments that can clearly have a cause-to-effect relationship (see comment 2, for example). This will add some value to the presented work. Also, it will be nice to validate previously published relationships between inorganic-ion/hormone concentrations and observed phenotypes. Are the expected phenotypes observed or not (comment 3)?

comment 1:

2.1.1: Even though tomato rootstocks of high Ca supply could be empirically selected (line 164), authors do not comment on why the same rootstocks do not show Ca concentrations when eggplant scions are crafted on them. Actually, this behaviour can be seen in all the experiments presented in the paper. Is there an interspecific barrier that prevents high inorganic ion or hormone levels in eggplant scions? There is a mention of this fact in Discussion (lines 476-481) but the authors do not provide any possible explanation for this observation. They have to elaborate more on this aspect of the results.

comment 2:

line 234-235: So, is the correlation mentioned at refs. 62-65 observed in the results of the present work?

comment 3:

lines 296-299: A correlation is suggested between Mo levels plant hormone biosynthesis (refs. 72, 73). Are results presented here in agreement with previously published work?

comment 4:

lines 320-321: no data are presented for auxin and gibberellin.

comment 5:

Add a table with the different scion-rootstock combinations

comment 6:

Figure 1: A and B are not visible on the figure

Figures: a, b, c, d, e and f annotations for statistical significance should be specified in the legends and in the supplementary Table S1.

Color assignment description in the legend should be simplified

What is the meaning of the different red-color scale within the tomato-tomato combinations?

comment 7:

Lines 318-322: Fig. S1 shows results only for SA and not for auxin or gibberellin. Also x-axis labels and error bars should be represented properly. Moreover, I can see that SA concentration is significantly higher than the controls in a lot of the tested combinations, contrary to author’s statement that SA concentration is not altered.

Round 2

Reviewer 3 Report

All my concerns have been addressed by the authors.